# Deep Brain Stimulation for Tremor: Update on Long-Term Outcomes, Target Considerations and Future Directions

**DOI:** 10.3390/jcm10163468

**Published:** 2021-08-05

**Authors:** Naomi I. Kremer, Rik W. J. Pauwels, Nicolò G. Pozzi, Florian Lange, Jonas Roothans, Jens Volkmann, Martin M. Reich

**Affiliations:** 1Department of Neurosurgery, University Medical Center Groningen, University of Groningen, 9713 GZ Groningen, The Netherlands; n.i.kremer@umcg.nl (N.I.K.); r.w.j.pauwels@umcg.nl (R.W.J.P.); 2Department of Neurology, University Hospital and Julius-Maximilian-University, 97080 Wuerzburg, Germany; pozzi_n2@ukw.de (N.G.P.); lange_l@ukw.de (F.L.); roothans_j@ukw.de (J.R.); volkmann_j@ukw.de (J.V.)

**Keywords:** deep brain stimulation, tremor, essential tremor, Parkinson’s disease, outcomes, clinical approach, target considerations, future directions

## Abstract

Deep brain stimulation (DBS) of the thalamic ventral intermediate nucleus is one of the main advanced neurosurgical treatments for drug-resistant tremor. However, not every patient may be eligible for this procedure. Nowadays, various other functional neurosurgical procedures are available. In particular cases, radiofrequency thalamotomy, focused ultrasound and radiosurgery are proven alternatives to DBS. Besides, other DBS targets, such as the posterior subthalamic area (PSA) or the dentato-rubro-thalamic tract (DRT), may be appraised as well. In this review, the clinical characteristics and pathophysiology of tremor syndromes, as well as long-term outcomes of DBS in different targets, will be summarized. The effectiveness and safety of lesioning procedures will be discussed, and an evidence-based clinical treatment approach for patients with drug-resistant tremor will be presented. Lastly, the future directions in the treatment of severe tremor syndromes will be elaborated.

## 1. Introduction

Over the last three decades, surgical therapies have been established as treatment for specific movement disorders. Deep brain stimulation (DBS) is now considered an evidence-based, routine therapy for patients with selected neurological and psychiatric disorders. The first, and still one of the most common indications, is drug-resistant tremor. Frequent etiologies are Essential tremor (ET) and Parkinson’s disease (PD). However, not every patient suffering from tremor is a good candidate for surgery and will have the same long-term benefit. In fact, defined etiology, precise targeting and adequate programming are essential for therapeutic success. 

We will first present the clinical characteristics and summarize the pathophysiological concepts of tremor in ET and PD, then we will focus on patient selection criteria, target considerations, associated treatment outcomes and programming algorithms, and lastly, we will elaborate on other neuromodulative treatment options and future directions in tremor interventions.

Rare and uncommon tremor syndromes will not be covered in this review, and we would like to refer to the excellent work by Ramirez-Zamora and Okun [1]. 

## 2. Clinical Characteristics and Pathophysiology

Tremor is defined as an involuntary movement that is both rhythmic and oscillatory [2]. The clinical spectrum of tremors is broad, and their classification depends on the condition (either rest or action) that activates the tremor. Here, we describe the clinical characteristics of the different types of tremor. 

Resting tremor occurs in a body part that is not voluntary activated while being completely supported against gravity. The amplitude of tremor should increase during mental tasks, diminish or disappear during active movement of the limb, and reoccur after its termination [3,4]. Action tremor is any tremor occurring on voluntary contraction of a muscle and is subdivided in postural, isometric and kinetic tremor. Postural tremor is present during voluntary maintenance of position against gravity. Isometric tremor occurs as a result of isometric muscle contraction (i.e., muscular contraction without movement). Kinetic tremor occurs during any voluntary movement. Intention tremor, a subtype of kinetic tremor, is identified when tremor amplitude increases during visually guided movements towards a target. It is also characterized by fluctuations of movement velocity when approaching the target. 

Tremor can be physiologic in healthy people, but is considered pathologic if a patient’s functional ability is affected. Enhanced physiological tremor is usually an upper extremity action tremor, induced by stress, endocrine disorders, drugs or toxins [5]. If a suspected enhanced physiological tremor is not reversible by treatment of the underlying cause, essential tremor is the most common differential diagnosis [2]. 

Essential tremor is a monosymptomatic, predominant postural or kinetic tremor which usually slowly progresses over the years. The topographic distribution in ET shows hand tremor in 94%, head tremor in 33%, voice tremor in 16% and leg tremor in 12% of the patients. ET may be hereditary and about 50–70% of the patients improve with intake of alcohol [6,7,8]. Although postural tremor is the most frequent form of tremor in ET, about 15% of the patients also show intention tremor. In almost half of the patients, this intention tremor occurs together with subtle signs of cerebellar dysfunction like ataxia and movement overshoot [9]. These patients also may suffer from mild gait disturbances [10]. Since 2018, the Task Force on Tremor of the International Parkinson and Movement Disorder Society defines ET patients with additional neurological signs such as tandem gait disturbances, subtle dystonic posture and memory impairment as essential tremor plus syndrome [2].

The olivary neurons are likely to represent the central oscillator of ET [11], while the network closely linked to the generation of tremor is certainly the cortico-bulbo-cerebellothalamo–cortical network [12,13,14]. Cerebellar degeneration and GABAergic dysfunction presumably also play a role in the pathophysiology of ET [15,16,17].

Parkinson’s disease can present any kind of tremor. The pill-rolling, resting tremor is typical, but up to 60% of PD patients have different forms of action tremor, which can occur alone or together with resting tremor [18,19]. In PD, tremor must be accompanied at least by bradykinesia. A simplified subdivision of tremors in three types is commonly used to describe its clinical presentation in PD [20]. Classical Type I Parkinsonian tremor is a pure resting tremor within a frequency of 4–6 Hz, but in the early stage of the disease, higher frequencies can also be found. In Type II the resting tremor is combined with action tremor of the same frequency, while its amplitude typically increases with mental stress, contralateral movements or during gait. When a voluntary movement is initiated, the tremor is temporarily suppressed and reoccurs after a few seconds with the hands outstretched, also called re-emergent tremor [3,4]. Type III is an isolated action tremor. This form is rare and often misdiagnosed as ET variants; a relation of this form of tremor with rigidity seems possible [18]. Lastly, Type IV represents mixed resting and action tremor with distinct frequencies, where ET and PD could coexist.

In contrast to ET, parkinsonian tremor is supposed to emerge from an aberrant activity in the cerebello-thalamo-cortical circuit, possibly triggered by dopamine (and serotonin) loss in the basal ganglia [21,22,23,24], but the central oscillator remains unclear [11].

The differential diagnosis between PD and ET can be difficult, especially in their early stage [25]. It has been estimated that 20% of the subjects with ET are misdiagnosed as PD patients and vice versa [26]. A good clinical test to differentiate between ET and PD tremor is to assess the suppression of resting tremor during movement, in PD the tremor ceases or greatly subsides when voluntary movement is executed [27]. If diagnosis on clinical tests proves to be difficult, dopamine transporter (DAT) imaging is recommended. This technique is used to detect presynaptic dopaminergic deficits with a sensitivity of almost 100% [28,29]. A summary of the differential diagnostic criteria is reported in Table 1.

## 3. DBS for Essential Tremor

DBS is a well-established second-line treatment for ET, showing tremor improvements of more than 60% [30,31,32,33,34,35,36]. The typical candidate for DBS has medication refractory symptoms affecting quality of life and the ability to work [37]. In these patients, treatment with propranolol and primidone at maximum tolerated doses did not result in sufficient symptom alleviation. An additional indication for DBS might be communication difficulties due to voice tremor [38,39,40]. 

The ventral motor part of the thalamus (Vim) was the first target for tremor management with DBS, representing an adaptation of the classical thalamotomy procedures of the 1970s. Several studies showed that Vim-DBS has a profound benefit in ET, not only for extremity tremor [30,36,41,42,43,44,45], but also for midline tremor (face, tongue, voice and head) [31,34,46]. In particular, a study with a follow-up of 7 years reported a reduction of postural tremor from 2.5 ± 1.4 at baseline to 0.5 ± 1.0 after final evaluation, and a reduction of action tremor from baseline 3.5 ± 1.0 to 2.0 ± 1.8 after 7 years, as rated by the Essential Tremor Rating Scale (ETRS) [32,47]. Significant tremor improvements were achieved with both unilateral and bilateral procedures [30,36]. Thus far, the largest Vim-DBS patient cohort is described in a prospective, controlled multi-center study that proved the safety and efficacy of uni- and bilateral Vim-DBS. In 122 ET patients from 12 centers an improvement of 2.49 ± 0.96 in the target limb severity score was found at 365 days after Vim-DBS [43]. A recent meta-analysis involving 1714 ET patients showed a pooled improvement in tremor score of 61% at a mean follow up of 20 months [34]. Consistent improvement in midline and extremity tremor were found. Moreover, Mitchell and colleagues studied the impact of unilateral Vim-DBS on axial tremor [44]. They found axial tremor to improve by 58% at 90 days post-surgery and 65% at 180 days post-surgery, based on pooled axial subscores in the ETRS. Staged bilateral Vim-DBS provided additional axial tremor improvement, mostly in head tremor, but also added more adverse events. In addition to tremor reduction measured with tremor rating scales, a recent study showed quantitative evidence that Vim-DBS is effective at reducing tremor across upper- and lower extremities, head and chin, as measured with an accelerometer [48]. 

However, Vim-DBS efficiency reduces over time, with total tremor scores showing a 55–65% improvement at 6 months, 53–66% improvement at 12 months, 50% improvement at 2–3 years, 45–46% improvement at 4–5 years and 33–48% improvement at ≥6 years after DBS surgery [42,43,49,50,51]. In addition, after a 2 year follow-up, quality of life gradually decreased, while depression and anxiety levels increased [46]. It is suggested that either disease progression or tolerance to the DBS stimulation (i.e., habituation) underlie this phenomenon of gradual reduction of Vim-DBS efficiency [52].

An adverse event rate of 32% has been reported for Vim-DBS [43]. Stereotactic-related adverse effects (infectious (2.3%), hemorrhages (2.3%), seizure (<1%) and stroke (<1%)), hardware-related adverse effects (15%) and stimulation-related adverse effects (33% and 34% for unilateral and bilateral procedures, respectively) were reported. Interestingly, the number of persistent stimulation-related adverse effects (i.e., dysarthria, abnormal gait and dysphagia) was bigger in the unilateral implanted group (16%) with respect to bilaterally implanted patients (11%). This contrasts with previous reports, that showed higher risks in bilateral implantation [30]. In a recent meta-analysis, stimulation-induced adverse events were most prevalent in Vim-DBS (24%), followed by device-related (12%) and surgical-related (6%) adverse events. The most common stimulation induced side effects were dysarthria (11%), paresthesia (6%), hemiparesis (6%) and headache (7%) [34]. 

These studies, including a recent systematic review, provide level 2 evidence that Vim-DBS is effective in the treatment of tremor in ET (Figure 1) [53].

The caudal Zona incerta (cZi) is one of the other eligible targets for DBS in ET [54,55]. Interestingly, by moving the stereotactic target from the Vim to the below white matter (i.e., cZi), Fytagoridis et al. showed a greater tremor reduction (92%) with respect to Vim stimulation (60%) [56]. Postural tremor score at baseline improved from 2.4 ± 0.9 at baseline to 0.1 ± 0.2 at final evaluation (mean 48.5 months). Action tremor improved from a preoperative score 3.4 ± 1.0 to 0.5 ± 0.6 after 3.5 years. Moreover, this benefit was accompanied with a stable improvement in hand function (78%; 9.1 ± 3.2 at baseline to 2.0 ± 2.1 at follow-up) at the 3 and 5 years follow-up in all the 18 (bilaterally) implanted patients. Another study by Fytagoridis et al. consisting of 50 ET patients stimulated in the cZi showed similar improvements of 95% in hand tremor (6.0 ± 2.3 at baseline, 0.3 ± 0.8 at 1 year follow-up) and 81% in hand function (10.5 ± 3.9 at baseline, 2.0 ± 3.0 at 1 year follow-up), based on the ETRS [57]. Furthermore, Eisinger et al. showed both Vim and cZI to maintain tremor and hand function improvement in the 4 years following DBS. However, a gradual worsening in tremor scores was seen over time in patients stimulated in the cZi, suggesting Vim-DBS to provide better long-term outcomes [58]. Still, according to patient reported outcome, cZi-DBS might be considered superior to Vim-DBS [59]. Bot et al. combined both targets by aligning Vim and the posterior subthalamic area (PSA, corresponding to the cZi among others) in one single surgical trajectory for DBS and showed successful tremor control (grade 0 or grade 1) in 69% of the patients [60]. 

PSA-DBS seemed to be effective according to the acute monopolar review of the stimulation in 16 patients with ET; over 600 stimulation settings were tested and a better tremor control was proved with respect to Vim stimulation although a higher occurrence of stimulation-induced side effects [61]. We recently showed that the PSA is a safe-spot against delayed onset gait disturbances and proposed a method for direct anatomical targeting on axial T2-weighted MRI [62]. As described by Blomstedt et al. the posterior and medial border of the subthalamic nucleus (STN) at the level of the largest diameter of the red nucleus may be used as a reference [55]. On the basis of these encouraging but preliminary results, a double-blind, prospective, cross-over study on PSA-stimulation for ET was conducted [63]. This study showed PSA-DBS to be more efficient, compared to Vim-DBS, providing similar tremor control with lower stimulation amplitudes. No differences between adverse events were reported. Ramirez-Zamora et al. conducted a review of uncontrolled trials reporting on PSA-DBS for tremor and showed a significant difference in tremor reduction between PSA stimulation and Vim stimulation (79% and 50%, respectively, *p* < 0.001) across different etiologies [64]. 

Only a few studies assessed the rate of adverse effects in cZi/PSA surgery and found them to occur in 20–46%. The most commonly reported stimulation-related adverse effect was dysarthria (20–57%), followed by spasms (34%) and postural instability (16–29%) of the electrodes implanted in the cZi/PSA. In addition, oculomotor side effects and paresthesias have been described in 19% and 11–14%, respectively [54,59,65,66].

Despite the fact that these encouraging results are based on heterogeneous studies, PSA-DBS might be considered in the treatment of drug-resistant tremor (level 2 evidence). However, long-term effects of PSA stimulation are still unknown. 

In the last years, the broad availability of diffusion tensor imaging (DTI) in clinical practice enabled a novel method of stereotactic targeting. The MRI-depended visualization of fiber tracts is used for directly targeting the dentato-rubro-thalamic tract (DRT) [67,68]. Fenoy et al. tracked the DRT on preoperative imaging to directly target the Vim in 20 ET patients and compared these patients to a historical cohort of 20 patients who underwent DBS surgery using atlas-based coordinates [69]. Both groups showed equal improvement of around 70% in arm tremor amplitude, with mean postoperative tremor scores between 0.69 ± 0.5 and 1.18 ± 0.2. In two cases described by Morrison et al., tremor improved by 68% and 72% (based on the ETRS) after intersectional targeting of the Vim using DRT tractography [70]. Low et al. conducted a double-blind, randomized study in 34 tremor patients (PD and ET) that either received conventional DBS insertion or DTI guided DBS insertion [71]. DTI guided DBS insertion showed better and more stable tremor control and fewer adverse effects compared to conventional DBS insertion. This is most likely because the DTI guided method allowed closer placement of DBS leads to the center of the DRT. Indeed, other studies confirmed the relationship between tremor improvement and the distance to the DRT, showing better tremor control when DBS contacts were in closer proximity to the DRT [72,73]. 

DRT targeting looks promising, but currently, it does not have the same level of evidence compared to Vim and cZI/PSA stimulation (Figure 1). However, we believe future research should provide more insight on the effectiveness and side effects of DRT stimulation. 

There are several options in DBS programming for ET and the stimulation parameters are usually individualized according to the condition of the patient. The parameters are routinely selected trough monopolar review of all contacts [74]. The final stimulation always represents a trade-off between tremor control and stimulation-related side-effects. There are three major stimulation induced side-effects, which could be possibly overcome with reprogramming. 

The first is loss of tremor suppression due to tolerance in the chronic course of stimulation (i.e., habituation) [42,52,75,76]. An adaptation of the biological response of the stimulated neuronal network is hypothesized to underlie this phenomenon, which classically occurs months after the implantation. “DBS holidays” have been reported to improve habituation, but adherence might be difficult for patients [77]. Turning off the stimulation overnight is also thought to prevent this delay-onset problem; however, in some cases, habituation still occurred [78]. Alternating stimulation patterns could be a better solution; patients treated with alternating stimulation patterns on a weekly, but not on a daily basis, showed less habituation than patients treated with constant stimulation [79,80]. In the future, closed-loop systems, delivering stimulation on-demand using feedback from physiological measures, might prevent the occurrence of habituation.

The second major stimulation induced side-effect is disturbance of gait in bilateral implanted patients, possibly due to antidromic cerebellar activation [62]. Contarino et al. were able to delay the onset of gait ataxia by pausing the stimulation overnight and suggested to shape the field of stimulation more dorsally and/or to restrict the volume of tissue activated with a bipolar stimulation. However, the reduction of VTA was associated with less effectiveness on tremor control in four patients [81]. An alternative option that showed sustained benefit of gait ataxia without reduction in tremor control is the shortening of stimulation pulse duration to 30 µs [82]. This might be due to fiber selective neurostimulation, which is possible when anatomically distinct fibers pathways show different diameters and, therefore, different chronaxies (a measure of neuronal excitability) [83]. In a more recent study, pausing thalamic DBS reversed stimulation-induced gait disturbances. Changing to a shorter pulse width (40 µs) maintained this improvement, while DBS also effectively suppressed the tremor [84]. 

The third major stimulation induced side-effect is dysarthria. Interleaving stimulation can be used to reduce dysarthria. It describes the alternating use of flickering of different stimulation programs on the same DBS electrode; using two different contacts prevents wider horizontal current spread. In a small study, there was no difference in tremor control, while 4/6 patients reported subjective improvement of speech during interleaving stimulation [85].

In conclusion, the best management for ET is still a matter of discussion. DBS for drug-refractory tremor in ET has currently been assigned level 1C evidence (Figure 1) [86]. A randomized, double-blind study comparing stimulation of the Vim and PSA should give more insight on which target is most effective for the treatment of tremor (clinicaltrials.gov NCT03156517). 

## 4. DBS for Parkinson Tremor

In PD, tremor is present in up to 79% of the patients and often results in severe social impairment [87]. DBS is shown to be effective in treating this symptom also in drug-resistant cases. Indeed, it was in 1987 that Alim-Louis Benabid firstly reported the benefit of thalamic DBS on drug-resistant tremor in PD, thus paving the way for the use of DBS in PD [88].

The DBS treatment of PD involves other targets connected to the ventral thalamus, namely the internal pallidum (GPi) and subthalamic nucleus (STN). A large body of literature supports the overall improvement in quality of life, motor symptoms severity and drug-related motor complications (i.e., dyskinesias and wearing-off periods) in patients with PD and DBS [89,90]. Compared to best medical treatment alone, DBS of the STN or GPi combined with medication was superior in controlling drug-responsive motor symptoms [91].

Data from over 1000 patients with PD and STN-DBS showed improvements in motor symptoms (50%) and activities of daily living (52%) according to the Unified Parkinson’s Disease Rating Scale (UPDRS) [92,93]. Specifically for tremor, improvements between 70–75% at one year following surgery were reported, and these improvements remained stable at five years [91]. 

Despite the fact that STN-DBS is generally favored over GPi stimulation for treatment of tremor, tremor responds similarly to both, and both targets show equal improvements in motor functioning [94,95,96,97,98,99]. The success of STN implantation is probably related to the management of stimulation programming. In addition, STN-DBS allows more reduction in antiparkinsonian medication [97,100,101]. However, bilateral GPi-DBS has been proven to significantly reduce tremor with 32–80% after 1 year [102,103,104] and 70% after 5–6 years [105]. Additionally, unilateral GPi-DBS showed tremor reduction between 47–71% [103,106].

Nowadays, STN and GPi represent the preferred targets for tremor control in PD because of their effect on rigidity and bradykinesia. The choice between the two is based on individual consideration (e.g., pharmacological management, non-motor-symptoms), although evidence suggested a slight superiority of STN-DBS over GPi-DBS (Figure 1) [87]. 

In case of tremor-dominant PD patients, targeting of the Vim is still considered. Early multicenter trials showed an effective management of tremor in PD patients with Vim-DBS [41]. These results were corroborated by long-term continuation studies [51,107]. However, the effect of Vim-DBS on the other PD symptoms is marginal and Vim-DBS is currently considered only in selected tremor-dominant non-fluctuating slowly progressive drug-resistant patients (Figure 1) [108].

The cZI may serve as an alternative or concomitant DBS target for PD tremor. In a recent randomized controlled trial, bilateral cZI-DBS patients showed better motor functioning (47%) and striking tremor improvements (92%) based on the UPDRS compared to the patients receiving the best medical treatment [109]. cZI has even been suggested to be superior to STN-DBS in improving contralateral motor functioning; however, dopaminergic medication doses often cannot be decreased [110]. Others have combined cZI and pedunculopontine nucleus (PPN) DBS and showed 69% tremor reduction (from 9.3 at baseline to 2.9 after combined stimulation according to items 20 and 21 of the UPDRS-III) [111]. Yet, the additional benefit of PPN in improving motor functioning is questionable. 

DBS programming in PD can vary greatly, but the benefit on tremor always goes along with the improvement of the others motor symptoms. During DBS programming, the effect on some clinical symptoms is not instantly visible. It may take minutes to see an improvement of rigidity and bradykinesia, while axial symptoms may take hours or days to improve with STN-DBS. In contrast, the effect on tremor is directly visible with Vim-DBS or STN-DBS in most cases [112]. While tremor and rigidity are the most frequent clinical signs targeted for improvement during the first programming session, the optimization of DBS parameters is usually attained within 3 to 6 months during 4 to 5 programming sessions.

## 5. Other Surgical Approaches

Before the introduction of DBS in 1987, ablative procedures, such as thalamotomy and pallidotomy, were well accepted in the treatment of movement disorders [88]. Cryothalamotomy, chemothalamotomy, radiofrequency (RF) thalamotomy and high-frequency ultrasound were performed to treat movement disorders in the early days [113,114,115,116,117,118]. Surgical therapy for PD started in the early 1900s as reports of the improvement following stroke lesioning of the anterior thalamus appeared [113,119,120,121]. These observational studies fostered surgical lesioning techniques, which since the 1980s showed varying levels of efficacy in PD. With respect to tremor, the best clinical outcome was achieved through lesioning of the anterior thalamus, thus identifying the ventro-intermediate thalamic nucleus (Vim) as an anatomical spot for tremor control. However, since the introduction of levodopa in 1968, interest in ablative neurosurgery fell into decline. Years later, a resurgence of interest in lesioning procedures occurred after side effects of levodopa became known. 

### 5.1. Radiofrequency Thalamotomy

RF thalamotomy is based on heating of the tip of a monopolar or bipolar RF lesioning probe. Comparable to DBS, the lesioning probe needs to be inserted through a burr hole in the skull and penetrated through the brain to the target. The Vim is the target of choice in most patients with medication-refractory tremor [122]. Before creating a permanent lesion, the effect on tremor reduction and presence of undesirable side effects can be evaluated by stimulation of the RF probe. In the awake patient, the effect is instantly visible after lesioning of the Vim.

Several studies describe the results after RF thalamotomy. Most are retrospective and report improvement of tremor in 70–100% of patients [123,124,125,126,127,128]. Nagaseki et al. performed selective Vim thalamotomy in 16 ET patients and 27 PD patients. At follow-up almost all patients had satisfactory tremor relief [122]. Jankovic et al. performed stereotactic thalamotomy in 60 patients with medication-refractory tremor of different etiologies. In all patients, 80% had marked or moderate improvement of tremor (86% in PD tremor, 83% in ET) [129]. Surgical complications are also described in these studies, such as dysarthria, gait disturbances, hemiparesis and intracerebral hemorrhage. These complications are often transient, but more prevalent compared to DBS [130]. 

RF thalamotomy is usually performed unilaterally. Bilateral RF thalamotomies are known for a higher risk of complications compared to unilateral lesioning [122,131,132]. Despite the fact that this is based on older studies, bilateral lesioning of the thalamus is still considered to be controversial. Only one single randomized controlled trial has been conducted to compare RF thalamotomy and DBS in a cohort of 67 subjects with PD and drug-resistant tremor. The study showed that Vim-DBS equals thalamotomy efficacy (complete tremor suppression in 27/34 and 30/33 patients, respectively) while inducing less adverse effects, thus resulting in a greater functional improvement [133]. Since this study was conducted in 2000, no study compared these two procedures in a similar randomized controlled trial with use of modern techniques. 

Similar to Vim-DBS, RF thalamotomy for the treatment of drug-resistant ET is assigned a 1C-level of evidence, because thalamotomy and DBS are equally effective [86]. Despite this fact, we feel the added value of reversibility, post-surgical adjustability and the possibility of bilateral procedures makes DBS the preferred neurosurgical treatment for tremor [134,135]. 

### 5.2. Focused Ultrasound (FUS)

DBS represents the current surgical standard for drug-refractory tremor, but other novel surgical approaches are considered in specific conditions, such as focused ultrasound (FUS) thalamotomy. This approach is currently considered when tremor is not controlled with pharmacological treatment and restrictions to DBS are present, such as fear of surgery, chronic immune compromised status or profession at risk of physical interaction with DBS device. In addition to surgical risks, risk of hardware failure, battery replacements and multiple hospital visits for adjustment of settings apply to patients with DBS. MRI-guided focused ultrasound (MRgFUS) is incisionless and only requires the patient to recline in an MRI device.

In 1959, Meyers demonstrated the application of lesioning with focused ultrasound [117]. For many years, a craniotomy had to be performed prior to the procedure, as the skull absorbed the acoustic energy. Since the development of phased array transducers in the 1990s, it is possible to transfer the energy through the skull without the need of an incision.

MRgFUS thalamotomy is based on a technique that delivers ultrasonic energy to a specific area in the brain. The lesion can be created with millimetric precision, due to MR guidance, as the local temperature can be accurately monitored with MR-thermometry in real-time. Comparable to RF thalamotomy, the effect is directly visible after creating a permanent thermal lesion. 

In 2013, the first two pilot studies of ET treatment with MRgFUS were published. Four patients were included and showed 81% relief of tremor (based on the ETRS) at 3 months after the procedure [136]. The other pilot study reported 75% improvement of hand tremor (based on the ETRS) after 12 months in 15 patients [137]. 

In a randomized controlled trial, MRgFUS thalamotomy reduced hand tremor in 56 patients with moderate-to-severe essential tremor [138]. Total tremor scores improved by 47% (from 18.1 ± 4.8 to 9.6 ± 5.1), without reduction of ipsilateral tremor and only minimal improvement in axial tremors of the head, neck and voice. Adverse events associated with MRgFUS thalamotomy included gait disturbances in 36% of patients and paresthesias or numbness in 38%. These adverse events persisted at 12 months in 9% and 14% of patients, respectively. Based on retrospective studies, the rate of adverse events seems similar between DBS and MRgFUS [36,139].

These encouraging facts raised interest in performing this non-incisional treatment in patients with tremor-dominant PD. In a randomized controlled trial, ETRS tremor score in PD patients improved by 62% after MRgFUS thalamotomy and 22% after sham procedure [140]. In a clinical trial including ET, PD and ET-PD patients, all 30 subjects experienced tremor relief directly after the procedure and the effect sustained in 80% of patients after 6 months. All but one patient reported improved quality of life [141]. With the publication of level 1 evidence, MRgFUS may be considered a valid treatment option for drug-resistant ET patients with unilateral tremor.

Similar to RF thalamotomy, MRgFUS is not indicated for bilateral treatment of tremor [142]. However, the efficacy and safety of staged bilateral FUS treatment of ET is currently being studied (clinicaltrials.gov NCT03465761).

### 5.3. Radiosurgery

Another non-incisional neurosurgical treatment of tremor is stereotactic radiosurgery (SRS), based on Gamma Knife (GK) radiosurgery. This procedure uses radiation therapy to create a lesion in the brain. Usually a dose between 130 and 150 Gy is chosen to create a lesion at the location of the Vim [143,144]. In contrast to RF and MRgFUS thalamotomy, the effect of GK thalamotomy is not directly visible. It can take several months until an effect is noticeable. 

Reported tremor reduction after SRS is variable (level 4 evidence). There are no randomized controlled trials comparing GK thalamotomy to other neurosurgical treatment modalities for tremor. Lim et al. performed a blinded evaluation of 18 ET and PD patients with a mean follow-up of 19.2 months [145]. Two patients showed marked improvement of tremor with one of these patients developing a serious adverse event. The authors reported a significant change in activities of daily living, but not in other tremor rating scale items. The improvement was seen 6 to 12 months after the GK thalamotomy. In contrast to this study, Young et al. examined 158 patients (102 PD, 52 ET, 4 other tremor) after GK thalamotomy [146]. There was a significant improvement in PD patients with 88% completely or almost tremor free (mean follow-up 52.5 months). Additionally, 88% of ET patients remained tremor free after 4 years. Half of the patients with a tremor of other origin improved significantly. Complications were reported in three patients (one transient, two permanent but mild). Other studies suggest GK thalamotomy remains a well-tolerated treatment for tremor patients ineligible for invasive stereotactic neurosurgery because of advanced age or comorbidities [147,148,149]. For future studies, it is important to conduct a long follow-up period, since complications may occur months or years after SRS.

## 6. Clinical Advice on Treatment of Drug-Resistant Tremor

Based on the available literature, various options are at hand on the neurosurgical treatment of patients with severe tremor. Usually percentages or absolute tremor score reductions are compared in studies. From a patient perspective, a reduction of 50% could still be a tremor markedly interfering with activities of daily living. In our opinion, successful surgery should be defined as a tremor score 0 or 1 according to a validated tremor rating scale. Additionally, the time point of assessment is of utmost importance, specifically for lesioning techniques (e.g., MRgFUS), where effect on tremor reduction should outweigh the adverse effects in the long-term. An assessment time point at 6 or 12 months may even be confounded by the microlesion effect. Therefore, it may be difficult to select the best possible treatment for the individual patient. 

Here, we present our clinical advice on the treatment of patients with drug-resistant tremor based on current literature (Figure 2). Patients with bilateral tremor due to ET are good candidates for DBS. Despite the fact that Vim stimulation and PSA stimulation might provide comparable tremor reduction, we feel Vim-DBS is preferred over PSA-DBS. Side effects are more likely to appear in PSA-DBS when the electrode is placed off-target, due to its proximity to the internal capsule. Efficacy and safety of DRT stimulation are yet to be elucidated. 

Patients with tremor-dominant Parkinson’s disease may undergo DBS as well. Despite the fact that STN and GPi are equally viable, STN stimulation allows for significant dopaminergic medication reduction. Vim-DBS in these patients might be an alternative, but it will influence other symptoms of PD not as much. For patients with unilateral tremor due to ET or tremor-dominant PD there are multiple options. Our first choice would be unilateral Vim-DBS, because of its proven efficacy and adjustability. RF thalamotomy may be considered when patients have contra-indications for DBS or do not prefer an implanted device. RF thalamotomy efficacy is similar to Vim-DBS, but it has a higher risk on side-effects due to its irreversible nature. When patients have contra-indications for surgery, MRgFUS is a proven incisionless treatment for unilateral treatment. SRS may be considered an alternative, but the effect and side-effects may become visible after several months.

## 7. Future Directions

The field of DBS for tremor has been rapidly growing over the past decades. Impressive long-term outcomes have been reported. Still, there is room for improvement. 

The main goal of DBS surgery is to achieve maximal tremor suppression and minimal side-effects in each patient. Programming of stimulation parameters plays a crucial role in reaching this goal [150]. However, programming remains a rather time-consuming, empirical task, manually performed by a highly trained and experienced clinician. Besides, stimulation is currently being delivered in a chronic, constant fashion, without considering variations in the clinical state throughout the day or symptom fluctuations induced by medication. During continuous stimulation, stimulation-induced side effects can occur, which may be related to stimulation of physiological, rather than solely pathological, brain activity [151]. Additionally, continuous stimulation could induce loss of tremor suppression due to tolerance to the stimulation. In this section, we will elaborate on the future directions in DBS for tremor to improve these flaws.

### 7.1. Visually Based Programming

The development of models to estimate the DBS-related volume of activation (i.e., VTA) ensures a novel source of information for DBS programming in the last years and paves the way to visually based programming [152,153,154,155]. This visual-guided programming focuses on the anatomical structures around patient-specific lead position and facilitates the selection of the active contact. Recently, it has been shown that VTA software presents significant concordance with clinical data for selecting contacts [156]. A refined tuning of the stimulation is imaginable when retrospectively defined areas of good clinical outcome, sweet spots, would be available in such visualization software [153,157]. These sweet spots are determined by using the voxels within the VTA and the corresponding clinical outcome. Dembek et al. used this method to create stimulation maps of DBS effects on tremor suppression and stimulation-induced side-effects for target selection and clinical evaluation [61]. Recently, in a larger cohort of 275 PD patients, Boutet et al. defined the tremor suppression sweet spot and validated this spot using machine learning [158]. Another option to optimize DBS programming is to automatize it by using particle swarm algorithms, an approach developed to identify DBS electrode configurations and stimulation amplitudes generating the most efficient tissue activation [159]. 

A further step forward would be linking the local effect of stimulation with its influence on the network activity. This can be determined using either normative atlases of average brain connectivity calculated from large subject cohorts or individual connectivity data [157,160]. Al-Fatly et al. used normative brain connectomes to identify connectivity patterns that allowed for the prediction of individual tremor suppression in ET patients [161], while Horn et al. used normative brain connectomes to compute the connectivity profile of effective STN stimulation in PD patients [162]. Sooner or later, such connectivity profiles focusing on tremor will become available to predict tremor suppression in an individual, patient-specific fashion. Presumably, these connectivity maps could then provide a priori information about the expected DBS effects and ultimately, they might be able to serve as a useful tool for clinicians in programming DBS.

These prediction models may revolutionize DBS programming for tremor, advancing the current monopolar review approach to a visual based one, a potential recently described in dystonia patients [163]. Functional imaging is another modality possibly shifting the current programming approach. Recently, functional MRI (fMRI) has been proposed as a biomarker of clinical DBS response [164]. Boutet et al. used fMRI data to identify brain activity patterns associated with clinical benefits. Subsequently, these brain response patterns were used to train a machine learning algorithm to classify whether a given stimulation setting was to be considered clinically optimal, or could be improved. In the future, fMRI might also serve as a tool to assist DBS programming. However, none of those promising approaches for computer-assisted DBS programming have been tested in a prospective trial so far, which is absolutely necessary to ensure patient benefit. 

### 7.2. Coordinated Reset DBS

Coordinated Reset (CR) DBS is more focused on restoring the network disorder rather than suppressing its pathological characteristics. It has shown auspicious results in computational [165,166], as well as pre-clinical [167,168] and clinical Parkinson’s disease studies [169]. In CR DBS, intermittent, pseudo-randomized, brief, low-intensity and spatially distributed pulse trains are delivered to the intended target. This results in desynchronization of pathological oscillatory network activity, causing therapeutic benefit that is thought to last even after the stimulus is delivered [167]. The two key advantages of CR DBS are (1) current reduction leading to less risk of stimulation-induced side effects and less power consumption and (2) persisting desynchronizing effects beyond treatment delivery. CR DBS is a promising novel therapeutic alternative to conventional DBS. However, applicability of CR DBS in tremor has not been studied, and this should be the topic of future research. 

Another new interventional strategy that may, in the future, be integrated into tremor treatment is the phase-locked suppression of aberrant neural tremor oscillation. Lately, this has been tested via transcranial electrical stimulation of the cerebellum [170], but it would also be suitable for an implanted device. The mechanism behind this innovative strategy is the active disruption of tremor oscillation generation in the olivocerebellar loop, targeting the temporal coherence of the pathology, rather than masking tremor oscillation by desynchronizing firing patterns in the thalamocortical loop as with DBS. Schreglmann et al. showed that this strategy could effectively suppress tremor during and shortly after stimulation [170]. Nevertheless, future studies are needed to further study the tremor suppression characteristics and determine the safety profile.

### 7.3. Adaptive DBS

Traditionally, DBS systems deliver electrical pulses through the implanted electrodes in a continuous fashion. However, in tremor patients, symptoms fluctuate throughout the day, depending on various factors, such as motor performance, stress and medication. A DBS system working with feedback signals to stimulate only when necessary would be valuable. 

In the last years, there has been growing interest in the potential of these ‘closed-loop’ DBS devices [171,172]. Closed-loop or adaptive DBS (aDBS) modulates stimulation in response to the patients’ current state of pathological activity. aDBS employs biomarkers to receive information, derived from either internal sensors, such as electrocorticography (EcoG) and local field potentials (LFP) or external sensors, such as electromyography (EMG) or kinematic data via an accelerometer.

Different internal sensors have been studied to trigger tremor-specific stimulation. In ET, motor cortex signals, gathered via EcoG, have been used to increase or decrease DBS, based on tremor magnitude [173]. Recently, Ferleger et al. described a fully implanted aDBS system in two patients with chronically implanted EcoG strips over the hand portion of M1 and DBS leads in the ipsilateral Vim and showed 33.2% more effective tremor suppression compared to cDBS [172]. Additionally, neural activity recorded from the DBS lead could provide tremor-related information, and local field potentials (LFPs), such as evoked compound action potentials (ECAPs), can serve as potential feedback signals for aDBS [174]. In PD, the most prominent biomarker used for aDBS is STN beta-band (13–35 Hz) LFP power, since it correlates with the severity of bradykinesia and rigidity [175], but not with tremor [176,177]. In tremor, beta-oscillations are often reduced, questioning the usefulness of beta activity alone to drive aDBS [178]. Still, Piña-Fuentes et al. found a 60% reduction in tremor scores with beta-activity driven aDBS in PD patients, which was not different from cDBS [171]. High-frequency oscillation (HFO) in the STN have also been shown to be associated with parkinsonian rest tremor [179], ratifying HFOs potential to serve as a novel aDBS biomarker for tremor-dominant PD phenotypes.

aDBS biomarkers can also originate from external sensors. Surface EMG signals have been shown to be able to predict tremor onset and could thereby facilitate activation of stimulation before the tremor appeared [180]. Furthermore, Cagnan et al. used an external accelerometer to sense a particular tremor phase and subsequently locked thalamic DBS to this tremor phase [181]. Tremor suppression was achieved despite using substantially less energy than cDBS. In patients showing suppressive effects (4/6 patients), tremor suppression was 64% on average, which is comparable to chronic Vim-DBS, but not all subjects exhibited tremor suppression during stimulation (2/6 patients). Another study used amplitude-responsive aDBS in tremor-dominant PD patients, detecting tremor power by a smart watch [182]. aDBS reduced tremor power by just 37%, but this was realized with voltages 76% lower than those used for cDBS.

It is not inconceivable that, in the future, internal and external sensors will complement each other. Besides, future advances in LFP tremor detection might provide additional aDBS biomarkers to suppress tremor. However, current aDBS studies describe case series [171,172,173,174,175,176,177,178,179,180,181,182] (level 4 evidence), thus, future studies should confirm these results by investigating more patients. In addition, any prospective biomarker needs to be clinically compared to cDBS, since this is the present gold standard.

## 8. Summary

Drug-resistant tremor in ET and PD has different effective neurosurgical treatment options. Of these, DBS is the most commonly used. However, other alternative procedures and targets are valuable if patient-specific characteristics are taken into account. In this review, we provided a possible clinical approach for patients with severe tremor undergoing surgical treatment. Despite the fact that DBS is around for more than twenty years, research on the most effective approach is still ongoing. Promising novelties here are patients-specific connectivity maps and biomarkers such as fMRI to assist DBS programming and particle swarm or other optimization algorithms to automatize DBS programming. Other innovative techniques are adaptive DBS based on LFP recordings and coordinated reset DBS.

## Figures and Tables

**Figure 1 jcm-10-03468-f001:**
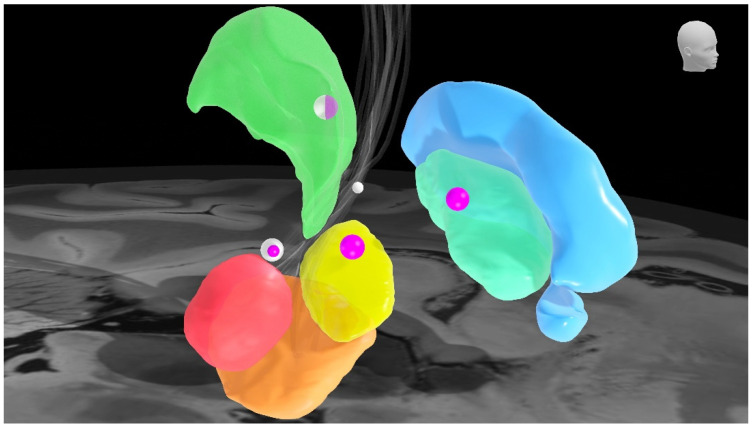
Deep brain stimulation targets and corresponding levels of evidence. The sphere size corresponds to the level of evidence for that specific target; higher evidence levels correspond to larger spheres. White spheres: essential tremor. Purple spheres: Parkinson’s disease. Brain nuclei: green: ventral motor part of the thalamus, sky blue: external pallidum, turquoise: internal pallidum, yellow: subthalamic nucleus, orange: substantia nigra, red: red nucleus. Grey fibers: dentato-rubro-thalamic tract.

**Figure 2 jcm-10-03468-f002:**
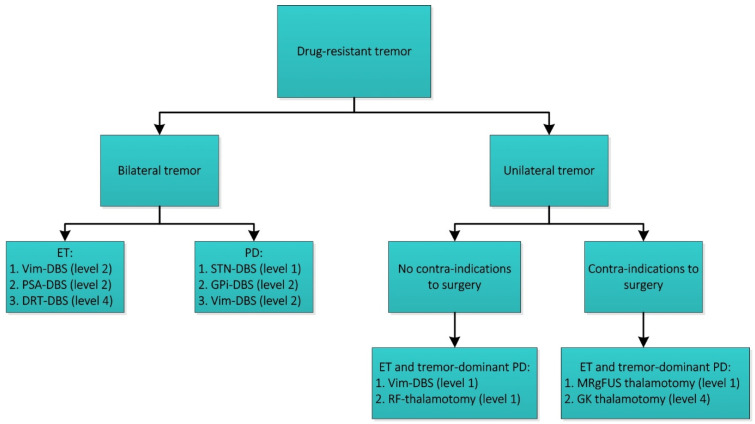
Surgical treatment options for patients affected by drug-resistant tremor. The levels correspond to the level of evidence according to the Oxford Centre for Evidence-Based Medicine 2011. DBS: deep brain stimulation, ET: essential tremor, PD: Parkinson’s disease, Vim: ventral intermediate nucleus of the thalamus, PSA: posterior subthalamic area, DRT: dentato-rubro-thalamic tract, STN: subthalamic nucleus, GPi: internal pallidum. RF-thalamotomy: radiofrequency thalamotomy. MRgFUS: MRI-guided focused ultrasound. GK: gamma knife (radiosurgery).

**Table 1 jcm-10-03468-t001:** Clinical criteria to separate essential and Parkinsonian tremor.

	Essential Tremor	Parkinsonian Tremor
Hereditary tremor	++	-
Head tremor	++	-
Voice tremor	++	-
Sensitivity to alcohol	++	-
Classical resting tremor	+	++
Predominant unilateral tremor	+	++
Leg tremor	+	++
Re-emergent tremor	-	++
Sensitivity to L-dopa	-	++

++: Likely to be present, + might be present, -: unlikely to be present.

## Data Availability

No new data were created or analyzed in this study. Data sharing is not applicable to this article.

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
