# Peer review of "Deep Brain Stimulation for Tremor: Update on Long-Term Outcomes, Target Considerations and Future Directions"

_jcm, 2021, doi:10.3390/jcm10163468_

Round 1

Reviewer 1 Report

This is an exhaustive and up-to-date review of neurosurgical treatment for tremors, with authors' recommendations for selection of treatment. I find the manuscript acceptable with only minor changes. A review like this will never be complete, but I find that the authors have found a representative selection of references. They also present good evidence for their treatment recommendations based on todays available litterature, but I am a bit surprised that they recommend RF-thalamotomy over MRgFUS for  unilateral tremor patients without contraindications to surgery (text p 11 and Fig 2). No containdications doesn't mean that you have to choose a surgical procedure! 

Some minor comments:

I suggest that the authors mark "Classical resting tremor" with +++ in Table 1. As I understand this table, it does not include PD patients without tremor, and classical resting tremor should then be very typical in PD.

In Fig 1 the colour of GPi looks light green or turquoise to me, not light blue as stated in  the text (and which is the colour in parts of GPe).

Line 379: "trail" should be "trial"

Reviewer 2 Report

Overall this is an excellent review paper and I fully endorse and support its publication as it will benefit its readers.

Few minor details that can be amended to make the paper flawless, but this should not be taken as a criticism of the work that the authors have done, which as I have said above I have rated as excellent.

  1. Line 152 (Page 4, paragraph 2): ‘…underly….’ Is an archaic word which in modern English is spelt as ‘underlie’.
  2. Line 265 (Page 6, paragraph 3) : ‘…..alternative flickering of….’ should be ‘alternating use of….’
  3. Line 284 (Page 6, paragraph 6): ‘Large literatures…’ should be ‘A large body of scientific literature supports…..’
  4. Lines 363 (Page 8, paragraph 2), Line 410 (Page 9, paragraph 5), Line 439 (Page 10, paragraph 2): the word ‘inflicting’ can be replaced by the word ‘creating’; ‘inflict’ can be replaced by ‘create’
  5. Figure 1 legend: the shades of blue depicted have more subtle names i.e. external pallidum is not ‘dark blue’ but is actually ‘sky blue’ and internal pallidum is not ‘light blue’ but is actually called ‘aquamarine’
  6. Line 378 (Page 9, paragraph 1), Line 472 (Page10, paragraph 5), Line 477 (Page 10, paragraph 6 ): ‘ Despite this is….’ should be ‘Despite the fact that this is….’; ‘Despite Vim stimulation….’ should be ‘Despite the fact that Vim stimulation…’; ‘Despite STN and GPi….’ should be ‘Despite the fact that STN and GPi……’
  7. Line 466 (Page 10, paragraph 4): ‘….outweight the adverse…on the long-term’ should be ‘…..outweigh the adverse…in the long-term’
  8. Line 468 (Page 10, paragraph 4): ‘……flawed by…..’ should be’….confounded by…..’
